# Recent Advances in Renal Tumors with TSC/mTOR Pathway Abnormalities in Patients with Tuberous Sclerosis Complex and in the Sporadic Setting

**DOI:** 10.3390/cancers15164043

**Published:** 2023-08-10

**Authors:** Payal Kapur, James Brugarolas, Kiril Trpkov

**Affiliations:** 1Department of Pathology, University of Texas Southwestern Medical Center, Dallas, TX 75390, USA; 2Department of Urology, University of Texas Southwestern Medical Center, Dallas, TX 75390, USA; 3Kidney Cancer Program at Simmons Comprehensive Cancer Center, University of Texas Southwestern Medical Center, Dallas, TX 75390, USA; 4Hematology-Oncology Division of Internal Medicine, University of Texas Southwestern Medical Center, Dallas, TX 75390, USA; 5Department of Pathology and Laboratory Medicine, Cumming School of Medicine, University of Calgary, Calgary, AB T2L 2K5, Canada; 6Alberta Precision Labs, Rockyview General Hospital, 7007 14 St., Calgary, AB T2V 1P9, Canada

**Keywords:** kidney cancer, tuberin, hamartin, PTEN, ccRCC, clear cell renal cell carcinoma, TFEB, TFE3

## Abstract

**Simple Summary:**

In the past decade, several novel renal neoplasms characterized by mutations in the tuberous sclerosis complex (TSC) or mechanistic target of rapamycin (mTOR) pathway genes in both the sporadic and germline settings have been described. Herein, we review these entities, highlighting their clinical and molecular characteristics.

**Abstract:**

A spectrum of renal tumors associated with frequent TSC/mTOR (tuberous sclerosis complex/mechanistic target of rapamycin) pathway gene alterations (in both the germline and sporadic settings) have recently been described. These include renal cell carcinoma with fibromyomatous stroma (RCC FMS), eosinophilic solid and cystic renal cell carcinoma (ESC RCC), eosinophilic vacuolated tumor (EVT), and low-grade oncocytic tumor (LOT). Most of these entities have characteristic morphologic and immunohistochemical features that enable their recognition without the need for molecular studies. In this report, we summarize recent advances and discuss their evolving complexity.

## 1. Introduction

Recent analyses of renal tumors in patients with tuberous sclerosis complex (TSC) syndrome have revealed a spectrum of morphologic subtypes which have subsequently been recognized in the sporadic setting [1,2]. Unlike other hereditary syndromes, such as von Hippel–Lindau disease (VHL) and hereditary leiomyomatosis and renal cell carcinoma (HLRCC), histologic manifestations of renal tumors associated with TSC/mTOR pathway alterations are heterogenous. However, characteristic morphologic and immunohistochemical (IHC) features allow their recognition without the need for molecular studies in many cases. The identification of these individual entities has improved diagnostic accuracy, and a subset previously regarded as “carcinomas” have been found to display benign behavior. Histologically similar sporadic counterparts reveal similar alterations in genes encoding components of the TSC/mTOR signaling axis. These tumors are characterized by the activation of the mechanistic (also called “mammalian”) target of rapamycin (mTOR) complex 1 (mTORC1), a master regulator of cell growth, metabolism, and autophagy [3]. Herein, we discuss renal tumors with frequent TSC/mTOR pathway gene alterations in both the germline and sporadic settings.

## 2. Tuberous Sclerosis Complex Syndrome

Tuberous sclerosis complex (TSC) syndrome (OMIM 191100) is a multi-organ syndrome characterized by hamartomas and benign tumors in various organs [4]. Clinical diagnosis is made based on two major diagnostic features or a combination of one major and at least two minor diagnostic features [5,6]. The major diagnostic criteria include: (1) subependymal giant cell astrocytoma; (2) cardiac rhabdomyoma; (3) lymphangioleiomyomatosis; (4) hypomelanotic macules (≥3 of ≥5 mm diameter); (5) angiofibromas (≥3) or fibrous cephalic plaques; (6) ungual fibromas (≥2); (7) shagreen patches; (8) multiple retinal hamartomas; (9) multiple cortical tubers and/or radial migration lines; (10) subependymal nodules (≥2); and (11) angiomyolipomas (AMLs) (≥2). Minor features include: (1) multiple renal cysts; (2) numerous hypopigmented macules; (3) dental enamel pits (>3); (4) intraoral fibromas (≥2); (5) nonrenal hamartomas; and (6) retinal achromic patch [4,7].

TSC is inherited with an autosomal dominant pattern and is estimated to affect ~2 million people worldwide (1 in 6000–10,000 new births) [8,9]. It is caused by germline loss of function (LOF) mutations in either *TSC1* (encoding hamartin; located at 9q34) or *TSC2* (encoding tuberin; located at 16p13.3) [10,11]. Genetic testing of “normal” tissue is often used to confirm the diagnosis of TSC. Approximately 60–70% of TSC patients have de novo germline mutations [12]. Observed in around 70% of TSC patients, *TSC2* germline mutations are typically associated with greater disease burden and severity than *TSC1* mutations [13,14,15]. This may be because TSC2, which binds TSC1 to form a complex, acts as the catalytic subunit while TSC1 stabilizes the complex. In ~10% of patients, no mutations in *TSC1* or *TSC2* are identified using standard sequencing tools. This may partly be due to mosaicism. Therefore, it is recommended that the diagnosis of TSC be primarily established using clinical criteria [5].

Disease penetrance is over 95%, but there is significant variability in severity and phenotype. Some patients may have subtle or no clinical manifestations, in part owing to the stochastic timing and frequency of the second hit [16,17]. Nevertheless, the loss of a single allele of *TSC1* or *TSC2* may be sufficient to cause some neurologic abnormalities and possibly other TSC-associated features. However, renal tumors require somatic inactivation of the remaining *TSC1* or *TSC2* allele [18,19].

## 3. Renal Tumors with TSC/mTOR Pathway Gene Alterations and Their Sporadic Counterparts

Kidneys are affected in 80–85% of TSC patients, and this begins as early as childhood [2]. The most common renal manifestations include cysts and AMLs, which are typically multifocal and bilateral, and rarely other renal tumors [17]. Renal cysts and AMLs are evident in up to 80% of children with TSC by the age of 10 [20,21]. Current recommendations include abdominal imaging at the time of diagnosis, which has led to the identification of more than 80% of AMLs prior to symptoms [22]. Patients with large AMLs may present with acute hemorrhage, flank pain, and hypotension from a ruptured AML [23]. Renal dysfunction is a common cause of morbidity and mortality in TSC patients due to disease burden, complications from ruptured AMLs (including acute hemorrhage), and as a consequence of interventions such as surgery and embolization [24].

In addition, 2–5% of TSC patients have a contiguous germline deletion involving both the *TSC2* and the adjacent *PKD1* (Polycystin 1) gene (also located on chromosome 16p13). This results in a more severe phenotype of a polycystic kidney disease, which often presents with renal insufficiency [25,26]. As for patients with TSC syndrome, patients with contiguous gene deletions may develop a wide spectrum of renal tumors [27].

Renal cell carcinomas (RCCs) develop in TSC patients at rates higher than in the general population (2–4%) [7]. Further, in TSC syndrome patients, RCCs often occur at a younger age and may affect children. In a subset of patients, multiple and/or bilateral tumors develop [12]. Most renal tumors in TSC patients are small, and they are often detected using routine surveillance imaging [2]. Median age at diagnosis is ~40, and there is a predilection for females [28]. The exact reason for this female predilection is not understood, but it may be related to estrogen. Renal tumors in this setting require a second-hit mutation [12]. Initially, these tumors were regarded as conventional clear cell RCCs [28,29], though they lacked the canonical chromosome 3p loss or *VHL* mutations [30].

The last decade has witnessed a growing interest in renal tumors in TSC patients. An early study by Schreiner et al. [31] reported on a 43-year-old man with TSC and bilateral renal lesions, including multiple AMLs, cortical cysts, and four separate “RCCs of unclassified type”. These tumors shared distinct morphology, including sheet-like, glandular, trabecular, or cystic architecture and abundant granular eosinophilic cytoplasm. Subsequently, in 2014, two groups laid the foundation for a new classification. Guo et al. [1] described 57 RCCs from 18 TSC patients and classified them into 3 morphologic groups: (1) chromophobe-like RCCs (59%); (2) renal angiomyoadenomatous tumor (RAT)-like RCCs (30%); and (3) eosinophilic/macrocystic RCCs (11%). Yang et al. [2] reported on 46 tumors from 19 TSC patients and classified them into 3 distinct morphologic groups: (1) TSC-associated papillary RCCs (52%); (2) hybrid oncocytic chromophobe tumors (33%); and (3) unclassified RCCs (15%). While the nomenclature in these two studies differed, they reported similar morphologies and IHC patterns. This led to multiple publications in the following years which characterized similar tumors in the sporadic setting with TSC/mTOR pathway mutations. We first described *TSC* gene mutations in RCC in 2011 [32,33], but these mutations are not unique to a particular RCC subtype, and the newly described entities lack other frequently mutated genes. Below, we summarize these entities using the terminologies endorsed by the GUPS (Genitourinary Pathology Society) [34] and by the recent 2022 World Health Organization (WHO) classification (5th edition) [35].

### 3.1. Angiomyolipoma and Variants

Renal angiomyolipoma (AML), a benign mesenchymal tumor, is characterized by the proliferation of perivascular epithelioid cells and is regarded as a PEComa. It represents the most common tumor in TSC patients (86%) [24]. The median age at diagnosis is 16.9 years [22]. In TSC patients, multiple and bilateral AMLs are often found [23]. Tumors can become bulky and merge, making it difficult to distinguish clear boundaries between individual lesions. Hemorrhage is concerning, particularly with actively growing AMLs that are >3 cm in diameter [28,36]. Patients with de novo disease or *TSC2* mutations may be at higher risk for AMLs and renal cysts [23]. However, AMLs are still more common in the sporadic setting, where they similarly exhibit biallelic loss of *TSC2* or *TSC1* [22,37,38].

The majority of AMLs have a triphasic morphology consisting of haphazardly arranged vascular, smooth muscle, and adipose tissue elements. AMLs almost always stain positive for classic melanocytic markers, including HMB-45 and Melan-A, as well as lysosomal markers, such as cathepsin K. Conversely, they uniformly lack markers of the renal tubular epithelium, such as PAX8 and keratins.

An awareness of the spectrum of AML histologies, such as “fat-poor” AML, angiomyolipoma with entrapped epithelial cysts (AMLEC), sclerosing AML, and AML with oncocytic features, can prevent misdiagnosis [24,39,40,41] (Figure 1). Epithelioid AML (epithelioid PEComa) is a rare variant composed of ≥80% epithelioid cells (Figure 1). Epithelioid AML may show two morphologic patterns: a carcinoma-like pattern of large atypical eosinophilic (ganglion-like) cells, or a diffuse pattern of plump spindle cells. Both show positivity for melanocytic markers and cathepsin K. Aggressive behavior in epithelioid AML is more frequent in the context of TSC syndrome. Other indicators of aggressiveness include a tumor diameter >7 cm, a carcinoma-like pattern, necrosis, and renal vein or perinephric fat involvement. Some epithelioid AML studies have reported distant metastases in up to one third of cases, particularly when >7 cm [42]. However, recent data from large longitudinal series suggest that the frequency of metastasis may be much lower (5%) [40].

### 3.2. Renal Cell Carcinoma with Fibromyomatous Stroma (RCC FMS)

Historically, RCC FMS has been described using different terminologies [43,44,45,46]. The name “renal cell carcinoma with fibromyomatous stroma” (RCC FMS) was officially endorsed by the GUPS in 2021 [34]. RCC FMS is commonly observed in the sporadic setting, but morphologically, immunohistochemically, and molecularly identical tumors have also been described as the most common RCC in TSC patients [1,2,24,41]. In the sporadic setting, RCC FMS is often associated with mutations involving the *TSC1*/*TSC2* or *MTOR* genes [24,34,47].

Macroscopically, these tumors, which can also be multifocal [48], are small, solid, tan-brown, and lobulated [43,49,50]. Microscopically, they have two prominent components: (1) an epithelial component consisting of tumor nodules with elongated and branched tubules lined by clear cells with voluminous cytoplasm, and (2) a prominent fibromuscular stroma (Figure 2). The nuclei are typically WHO/ISUP grade 2–3. The fibromyomatous stroma is often prominent at the periphery, but it can also separate the clear cell nodules [34,43,49].

Immunohistochemically, RCC FMS is characterized by diffuse CAIX and cytokeratin (CK)7 as well as focal CD10 expression, and is negative for AMACR [34,47,51]. Other positive markers include vimentin and high molecular weight cytokeratin. Some have also reported apical staining for CK20 [24]. CAIX staining is usually diffuse membranous, though focal “cup-shaped” staining, similar to that seen with clear cell papillary renal tumors (ccPRTs), can also be observed. While ccPRTs show similarly diffuse CK7 staining along with “cup-shaped” CAIX staining, they more characteristically have apically oriented low-grade nuclei and scant clear cytoplasm. Further, ccPRTs are positive for GATA3 and negative for CD10 (and AMACR). The differential diagnosis for RCC FMS associated with TSC/mTOR mutations includes *ELOC*-mutated RCC, which are indistinguishable by their morphology and IHC characteristics, but show *ELOC* mutations and monosomy 8 [52], and a subset of ccRCCs that have more prominent fibromyomatous stroma.

The prognosis is generally favorable, and the majority of cases have an indolent clinical course [34,47,53]. However, three TSC patients with RCC FMS were reported to have developed lymph node metastases [1,2,24,27].

### 3.3. Eosinophilic Solid and Cystic Renal Cell Carcinoma (ESC RCC)

ESC RCC is the current name for “eosinophilic/macrocystic RCC”, which was initially described in TSC patients [1,31]. Since the first report, over 60 cases have been described in the literature. ESC RCC have been found more frequently in the sporadic setting, with a frequency of ~0.2% [54]. They occur across a broad age range, including in children, and are particularly common in female patients with TSC [54,55,56]. Though most ESC RCCs have indolent behavior, rare cases with metastatic disease have been reported [57,58,59]. ESC RCCs typically demonstrate biallelic loss of *TSC2* or *TSC1,* resulting in the activation of mTORC1 [57,60,61,62,63].

ESC RCC exhibit identical morphologic and IHC patterns in the sporadic and germline settings [48]. Characteristically, ESC RCC have grossly identifiable solid and cystic components (rare predominantly solid tumors showing only rare microcysts have been reported). Most tumors are less than 5 cm in diameter [54,55]. Microscopically, the cysts are lined by cells with hobnailing. The solid components are typically composed of cells with voluminous eosinophilic cytoplasm arranged in diffuse, compact, acinar, or nested patterns [54,55]. Scattered psammoma bodies, foamy histiocytes, and lymphocytes are common. The presence of coarse basophilic to purple cytoplasmic granules (“stippling” and “leishmaniasis-like” bodies) is a diagnostically helpful feature (Figure 2). Electron microscopy has revealed that these granules likely represent aggregates of rough endoplasmic reticulum and granular material [54]. Melanin pigment was recently reported in one ESC RCC [64].

Immunohistochemically, ESC RCC are reactive for CK20, cathepsin K, and vimentin. CK20 and cathepsin K expression is often focal and in rare cases negative. Negative markers include CD117 (KIT) and CK7 [54,55,65].

Though the morphology is often classic and molecular studies are not needed to make the diagnosis, the morphology and IHC patterns may sometimes overlap with those of epithelioid AML [59] and TFEB-altered RCC. In this context, a PAX8 IHC and fluorescence in-situ hybridization (FISH) for *TFEB* may aid in the diagnosis [65,66].

### 3.4. Eosinophilic Vacuolated Tumor (EVT)

EVT was first described by He et al. [67] as a “high-grade oncocytic tumor (HOT)” and by Chen et al. [68] as “sporadic RCC with eosinophilic and vacuolated cytoplasm”. EVT emerged from a group of tumors that share features with renal oncocytoma and chromophobe RCC (ChRCC) [69,70,71]. Tumors with similar histomorphology were also reported in patients with TSC [1,24,48,72]. The term “EVT” was proposed by the GUPS [34] and is now preferred [34,73]. EVT is found in patients of a broad age range, and occurs more frequently in women [34,67,68,74]. About 50 EVTs have been reported to date, and all cases have had benign behavior [34,74,75].

In the sporadic setting, EVT is typically solitary and solid, relatively small (<4 cm), and tan to brown [34,67,68,74,76]. Microscopically, the tumor cells are arranged in compact nested and tubulocystic patterns. In some cases, a more prominent loose stromal component can be intermixed. The cells have abundant eosinophilic cytoplasm and prominent intracytoplasmic vacuoles. The nuclei are enlarged, round to oval, with prominent nucleoli that focally resemble viral inclusions [67,68] (Figure 2). Thick-walled vessels are virtually always present at the periphery, but a well-formed capsule is lacking.

Immunohistochemically, EVTs are reactive for CD117, CD10, and cathepsin K. They typically exhibit only rare scattered cells positive for CK7 and CK20 [67,74]. It has been revealed via electron microscopy that EVTs demonstrate numerous intracytoplasmic mitochondria as well as dilated cisterns of rough endoplasmic reticulum [75,76].

Non-overlapping mutations in *MTOR*, *TSC2*, or *TSC1* are consistently found in EVTs, and these result in mTORC1 pathway activation, as shown, for example, by phospho-S6 and phospho-4EBP1 staining [68,76]. In tumors with *MTOR* mutations, the loss of the wild type allele on chromosome 1 (where *MTOR* resides) has consistently been observed [68,74,76]. Isolated losses of chromosomes 19p, 16p11, and 7q31 have also been reported [67,74]. Interestingly, there is morphologic and immunohistochemical overlap between EVTs and tumors with *FLCN* mutations (which, when in the germline, cause Birt–Hogg–Dube (BHD) syndrome), although *FLCN*-associated cases tend to have mosaic patterns and lack high-grade nucleoli [74].

### 3.5. Low-Grade Oncocytic Tumor (LOT)

LOT is another recently described entity that shares features with renal oncocytoma and ChRCC [43,77]. Though common in the sporadic setting, identical tumors have also been reported in patients with TSC [1,24,48,78,79,80]. LOT can also occur in end-stage kidney disease [79]. These tumors are usually observed in older individuals, and there is no gender predilection. No metastases have been reported to date in the over 100 cases described [34,77,78,79,81,82,83,84].

LOT is usually solid with tan-yellow to mahogany-brown cut surface (similar to oncocytoma) without necrosis or cysts [77,79,81]. LOT lacks a well-formed capsule and has a diffuse and solid growth pattern [34,77]. The neoplastic cells are eosinophilic with round to oval “low-grade” nuclei that lack irregularities but have frequent perinuclear clearing (halos), at least focally (Figure 2). Sharply delineated, edematous stromal areas with scattered individual cells or irregular “tissue culture” cell arrangements are frequent [34,77]. Such areas often contain hemorrhage. Adverse features, such as coagulative necrosis, nuclear pleomorphism, multinucleation, and mitotic activity, are uniformly absent.

LOT cells are diffusely reactive for CK7, but are negative for CD117, CD10, vimentin, cathepsin K, and CK20. In rare cases, focal and weak positivity for CD117 has been observed [77]. LOT is also consistently positive for GATA3 and exhibits focal phospho-S6 and phospho-4EBP1 staining [62,78,83]. As shown by electron microscopy, LOT exhibits abundant, closely packed cytoplasmic mitochondria, similar to oncocytoma [75].

The frequent involvement of mTOR pathway genes has been reported in LOT [78,83,85,86,87]. FOXI1, which is typically expressed in oncocytoma and ChRCC (and characterizes the normal intercalated cells of the distal renal tubules) is negative in LOT [41,88,89]. Further, chromosomal 1 loss, or *CCND1* rearrangements (as observed in oncocytomas) are not found in LOT [79].

## 4. Convergence on Mechanistic Target of Rapamycin (mTOR) Complex 1 Pathway

*TSC1* and *TSC2* encode hamartin and tuberin, respectively. These proteins interact to form a heterotrimeric complex (along with TBC1D7) [3]. This complex functions as a GTPase activating protein (GAP) and maintains RHEB, a small GTPase, in its GDP-bound form [90]. In its GTP-bound form, RHEB activates mTORC1. Thus, the TSC complex normally acts to restrain mTORC1 under conditions that are adverse for proliferation, such as when growth factors are absent [3] or when oxygen is scarce [91].

mTOR also integrates signals from nutrients, in particular amino acids. In situations of abundance, mTORC1 is recruited to the surface of the lysosome, where it is activated by RHEB [92]. mTORC1 recruitment is dependent on the RagGTPase complex. The RagGTPase complex is a heterodimeric complex composed of RagA (or RagB) and Rag C (or RagD). In its “active state”, RagA/B is bound to GTP and RagC/D is bound to GDP. Thus, proteins that regulate the nucleotide state of the dimer, such as the FLCN/FNIP (either FNIP1 or FNIP2), which functions as a GAP towards RagC/D, relay signals from amino acids to mTORC1. By stimulating the GTPase activity of RagC/D, FLCN/FNIP contributes to mTORC1 activation.

mTOR is a serine–threonine kinase of the phosphatidylinositol 3-kinase (PI3K)-related kinase family. mTOR forms a dimer that is positioned in a relatively open state when inactive (apo state). Upon RHEB binding, the dimer rotates to align active site residues for catalysis [93,94]. Cancer-associated mTOR mutations occur at specific residues and often destabilize the apo state, leading to kinase hyperactivation [93]. mTOR forms the catalytic subunit of two protein complexes: mTOR complex 1 (mTORC1) and mTOR complex 2 (mTORC2). mTORC1 is inhibited by rapamycin [95]. mTORC1 phosphorylates p70S6 Kinase 1 (S6K1) and eIF4E Binding Protein 1 (4EBP1) [3]. mTORC1 activation leads to cell growth, increasing the production of proteins, lipids, and nucleotides and suppressing catabolic pathways, such as autophagy [3] (Figure 3). mTORC1 also induces HIF-1α and angiogenesis, thereby coupling cell growth with an increase in oxygen and nutrient delivery [86,96]. In addition, mTORC1 suppresses protein catabolism and autophagy. mTORC1 inhibits both the early and late phase of autophagy through the phosphorylation of ULK1, ATG4, and UVRAG [92]. In addition, mTORC1 inhibits the transcription factor EB (TFEB), a master regulator of lysosome biogenesis and autophagy [97]. Specifically, mTORC1 phosphorylates three N-terminal residues S122, S142, and S211, which cooperate in excluding TFEB from the nucleus, at least in part through 14-3-3 binding, thereby preventing the activation of target genes [98,99]. mTORC2 instead phosphorylates Akt as well as PKCα and has been implicated in cytoskeletal regulation [92].

Interestingly, TFEB regulation in TSC-deficient cells is paradoxical. TSC-deficient cells are characterized by overactive mTORC1, which would be expected to phosphorylate TFEB in its N-terminus and lead to cytoplasmic sequestration and its inactivation. In contrast, we discovered that TFEB is constitutively nuclear and active in TSC-deficient cells [100]. Molecular and structural biology studies have shed light into this conundrum. Structural analyses have revealed that TFEB is phosphorylated by mTORC1 through an unconventional mechanism (one that is different from the canonical mechanism involving 4EBP1 and S6K1 targets). In contrast to canonical substrates, TFEB is recruited to the mTORC1 complex through the Rag GTPase complex. Specifically, RagC-GDP binds TFEB and positions it so that it can be phosphorylated by mTORC1 [101]. Interestingly, in TSC-deficient cells, low levels of RagC-GDP (likely the result of a dysfunctional FLCN/FNIP GAP complex) prevent TFEB1 recruitment to mTORC1, and thereby prevent its phosphorylation [102,103,104]. Thus, in TSC-deficient cells, both mTORC1 and TFEB are constitutively active [100,102,103]. These data raise the possibility that TFEB may be an important effector of tumorigenesis in TSC-deficient tumors. Several lines of evidence support this notion. First, TFEB accounts for a significant percentage of gene expression changes associated with TSC loss and mTORC1 activation (~25%) [100]. Second, rapalogs, which have been shown to be active against tumors in TSC patients and animal models, inhibit both mTORC1 and TFEB, which is then excluded from the nucleus [100,102,103] (Figure 4). Third, TFEB depletion reduces tumor formation in TSC-deficient xenografts [102,103]. Fourth, mutations in *FLCN*, which similarly disrupt the FLCN/FNIP GAP complex, result in renal tumors, which can be abrogated by the simultaneous inactivation of TFEB [104]. Fifth, in PEComas, a lack of TSC mutations may be offset by mutually exclusive *TFE3* translocations with constitutive TFE3 activation [105]. Finally, the finding that cathepsin K, a lysosome protease, is induced in some TSC/mTOR-deregulated tumors suggests that TFEB (or possibly TFE3), which is a master regulator of the lysosome, may also be activated. In relation to this, it is also worth noting that GPNMB (glycoprotein nonmetastatic B), a TFE3/TFEB target gene, is significantly upregulated in TSC/mTOR-mutated renal tumors (such as ESC RCC, LOT, AML, and PEComa), and that the levels are comparable to those found in translocation RCC, a tumor type driven by constitutive TFE3 (or TFEB) activation [106].

## 5. Clinical Management and Implications

Recognition of the spectrum of TSC-related renal tumors highlights the pivotal role pathologists play in ascertaining prognoses and directing the need for subsequent therapies, which hinge upon accurate pathological lesion identification. It is also not infrequent for the first diagnosis of a family of hitherto unrecognized hereditary renal neoplasia to be initiated by a pathologist. The National Comprehensive Cancer Network guidelines now recommend genetic risk evaluation for patients with (1) a diagnosis of renal tumor at an age ≤ 46 years; (2) bilateral or multifocal tumors; and (3) a first-degree relative with RCC. Genetic testing is particularly fruitful in patients with non-clear cell RCC [107]. Similarly, specific pathologic findings, such as succinate dehydrogenase (SDH) deficiency or BHD (resulting from germline *FLCN* mutations) related tumor histology, should also initiate germline testing.

For TSC patients, the current guidelines recommend baseline abdominal imaging at the time of diagnosis, followed by lifelong assessment every 1–3 years to monitor the development or progression of AMLs or cystic renal disease [6]. Renal AMLs can easily be identified via imaging due to the presence of fat. Magnetic resonance imaging (MRI) is the preferred modality because it affords a better structural evaluation of the lesions, including the identification of fat-poor AMLs, it does not involve nephrotoxic contrast, and it does not expose the patient to radiation. A biopsy is recommended for growing, non-fat-containing, solid tumors that pose diagnostic challenges on imaging.

The management of renal lesions depends on their histological type and size [108]. Classic AMLs can be safely observed while they are small. Bleeding or growing AMLs may be treated surgically or by embolization [6]. Epithelioid variants should be considered for earlier intervention given their aggressive behavior. In 2010, we and others showed that epithelioid AMLs respond to rapalogs [109,110]. mTORC1 inhibitors can also be effective in other settings. This was highlighted in our recent study of a TSC patient with multiple growing LOTs, where the patient benefited from mTORC1 inhibition, which forestalled the need for nephrectomy and dialysis [78]. Though many of the RCCs in TSC patients are indolent, some can occasionally metastasize, and their management can pose a challenge. The identification of small RCCs in the setting of multifocal AMLs requires careful surgical planning to minimize the loss of renal parenchyma. Thus, the treatment of TSC-associated RCCs often requires specialized, multi-disciplinary expertise, as was discussed in a recent review [111].

The recognition that TSC-associated tumors develop as a consequence of dysregulation of the mTORC1 pathway has paved the way for trials that have shown the therapeutic benefit of rapalogs. Results from the double-blind, placebo-controlled, EXIST-2 trial, which included 118 adult TSC patients, showed a >50% reduction in AML volume in 42% of the patients receiving everolimus (versus 0% of patients who received a placebo) at 8 months [112]. Response rates increased to 58% at 4 years. Similar results were seen in pediatric patients [113]. Based on the results from EXIST-2, everolimus was approved by the FDA as a first-line therapy for patients with AMLs > 3 cm in diameter. However, rapalogs are primarily cytostatic, and treatment discontinuation results in tumor regrowth (though at a slower rate), making lifelong therapy necessary [114].

## 6. *TSC*/*MTOR* Pathway Activated Renal Tumors—Lessons Learned and Future Directions

While remarkable progress has been made during the past decade in delineating *TSC*/*MTOR* mutation-related kidney tumors in both the hereditary and sporadic settings, several questions remain. An important question is whether these tumors represent distinct entities with different biological characteristics and prognoses, or whether they comprise a spectrum of histologies associated with mTOR pathway activation, for which a unifying terminology may suffice [84]. Based on current evidence, we do not think that entities such as ESC RCC, LOT, and EVT should be grouped as “*TSC/MTOR* mutation-associated renal tumors”. A recent study by Xia et al. [115] showed that ESC RCC, EVT, and LOT represent separate renal entities with consistent morphologic, immunohistochemical, and gene expression profiles, despite the fact that they all have *TSC/MTOR* abnormalities. While *TSC/MTOR* mutation-associated renal tumors share common pathway alterations, the type of mutation and the cell of origin may influence their biology and clinical behavior [115]. In addition, while they may share mutations in genes of the same pathway, there are also several distinguishing molecular features. For example, ESC RCCs have biallelic inactivating mutations (frequently frameshift and splice site mutations) abrogating TSC2 (or TSC1) protein expression [60,115]. In contrast, EVTs and LOTs often have activating mutations in *MTOR.* mTOR p.L2427 is particularly frequent in EVTs [68,74,76]. Further, mTOR-mutated EVTs typically lose the remaining wild-type *MTOR* copy through chromosome 1 loss [76,78]. In contrast, LOTs are typically heterozygous [78]. This is significant in particular since mTORC1 dimerizes and incorporation of wild-type subunit into a dimer with an activating mTOR mutation may reduce overall complex activity. Thus, mTOR pathway activation may not be as high in LOT as in EVT [78]. Finally, when present in EVT and LOT, *TSC1/2* mutations are frequently missense [115].

Indeed, *TSC* mutations (even if at low frequency) have been found in a broad spectrum of common (and uncommon) renal entities, which suggests that they are not characteristic of or specific to any particular subtype [32,116,117]. Further, as for other tumor suppressor genes, mutations in *TSC1/2* may not always represent driver events, in particular in the absence of heterozygosity and mTORC1 activation. However, in some RCC subtypes, somatic alterations in mTOR pathway genes occur in a background of classic mutations associated with the respective RCC subtype, and may drive tumor progression [32,61,89,116,117,118,119,120]. For example, we have shown in a genetically engineered mouse model of clear cell RCC with prototypical *VHL* and *PBRM1* mutations, that the addition of a mutation in *TSC1* results in higher-grade tumors [121]. However, how tumors are affected (at the phenotypic and molecular levels) by cooperating mutations is not always clear. This contrasts with the entities described in this paper, which otherwise have low mutation burden and lack other frequently mutated genes.

Further, the implications of TSC/mTOR pathway mutations in other sporadic tumors are less clear. For example, whereas mutations in TSC/mTOR pathway genes in the entities described herein are associated with responsiveness to rapalogs, this is less clear in the sporadic setting, where these mutations do not uniformly predict responsiveness [122]. Furthermore, while core tumorigenic pathways tend to be rewired, often through mutation, following the application of an antagonistic drug, this does not appear to be the case for sporadic RCC with TSC/mTOR pathway mutations [123]. This is in contrast to other tumor types, were rapalog administration results in the acquisition of mTOR resistance mutations [124]. It is notable in this context that while two rapalogs, temsirolimus and everolimus, are approved by regulatory agencies for the treatment of advanced RCC, to our knowledge, no resistance mutations have been reported to date [123]. This observation raises questions about the mechanism of action of rapalogs in RCC [125].

The rapidly expanding spectrum of *TSC/MTOR* mutation-associated renal tumors may pose diagnostic challenges for general pathologists given the rarity of some of the entities, the overlap in morphologic features, and the limited availability of some IHC stains. Furthermore, additional workup is required to distinguish RCC FMS associated with *TSC/MTOR* mutations from *ELOC*-mutated RCCs and ccRCCs with similar stromal features. There is also a possibility that diagnostic misclassifications may occur in contemporary practice, as some are still labelled as “carcinomas” (for example, “eosinophilic ChRCC” or “unclassified eosinophilic/oncocytic RCC”). These issues may be further compounded in limited needle biopsy specimens. It is also unclear whether these entities require different therapeutic approaches. However, as the great majority are benign or of low malignant potential, they can be managed conservatively, and complete resections are likely to be curative. Another challenge is that some *TSC/MTOR* mutation-associated renal tumors fail to cleanly fit a given morphological/IHC-defined category. For example, a recent study by Xia et al. [115] identified a group of entities that were not easily classifiable (“unclassified renal tumors with *TSC/MTOR* mutations”), even if they showed gene expression signatures that overlapped with ESC RCC. However, most of these renal tumors have distinct morphologic features and IHC findings that allow their diagnosis without the need for molecular studies (Table 1).

The recent Genitourinary Pathology Society (GUPS) consensus conference addressed the constellation of problems resulting from a lack of standardized terminology [43]. This should lead to better diagnostic characterization of these entities. Further studies in the upcoming years may help clarify the specific TSC/mTOR gene alternations and identify other factors (genetic, epigenetic, or host factors) that influence tumor development.

## Figures and Tables

**Figure 1 cancers-15-04043-f001:**
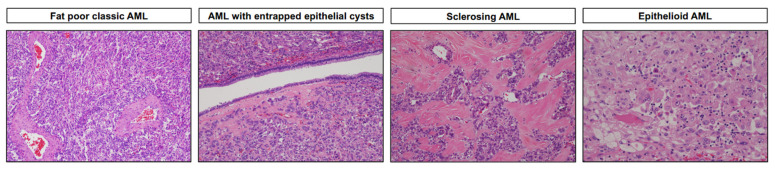
Representative hematoxylin and eosin-stained images depicting morphologic features of angiomyolipoma (AML) and its variants (magnification: 100× and 200×).

**Figure 2 cancers-15-04043-f002:**
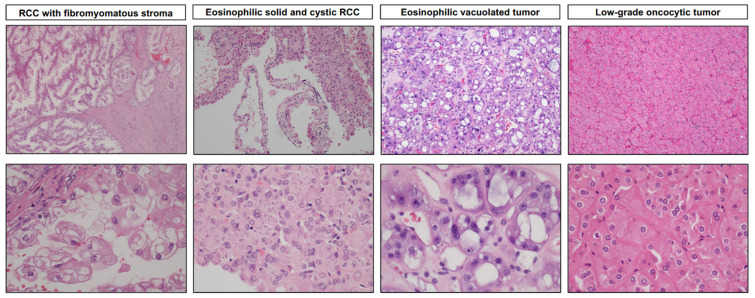
Representative microscopic images of recently described TSC/mTOR pathway gene-mutated renal entities. RCC with fibromyomatous stroma exhibits a nodular growth pattern, branching tubules, and papillary architecture featuring epithelial cells with clear to eosinophilic cytoplasm mixed with a fibromuscular stroma. Eosinophilic solid and cystic RCC exhibits solid and cystic components, with cells containing voluminous eosinophilic cytoplasm and characteristic coarse purple cytoplasmic granules (stippling). Eosinophilic vacuolated tumors exhibit a characteristic diffuse solid growth pattern, with tumor cells characterized by abundant eosinophilic cytoplasm and prominent intracytoplasmic vacuoles, large round nuclei, and prominent nucleoli. Low-grade oncocytic tumors show diffuse, solid, compact nests with neoplastic cells characterized by eosinophilic cytoplasm and round nuclei with focal perinuclear clearing (magnification: top: 40× and 100×; bottom: 400×).

**Figure 3 cancers-15-04043-f003:**
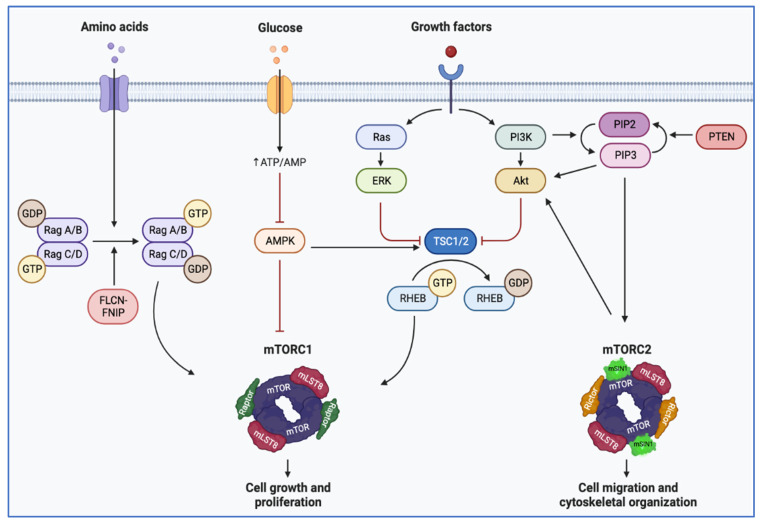
Illustration depicting mTOR pathway regulation. Extracellular signals from growth factors stimulate cell growth and proliferation. Downstream of receptor tyrosine kinases, the activation of PI3K leads to inactivation of the TSC1/TSC2 complex, which functions as a GTPase-activating protein and maintains RHEB in its inactive form. RHEB cooperates with Rag GTPases, which integrate signals from nutrients. Rag GTPases are anchored at the lysosome membrane, providing a docking site for mTORC1 by directly binding Raptor. When nutrients are plentiful, mTORC1 is docked at the surface of the lysosome by Rag GTPases and is activated. Amino acid scarcity switches the Rag heterodimer to an inactive state, resulting in the release of mTORC1 from the lysosome (created with BioRender.com). (AMPK: AMP-activated protein kinase; Akt: protein kinase B; FLCN: folliculin; FNIP: folliculin-interacting protein; GDT: guanosine diphosphate; GTP: guanosine triphosphate; mTORC1 (or 2): mammalian target of rapamycin complex 1 (or 2); PTEN: phosphatase and tensin homolog; PIP2: Phosphatidylinositol 4,5 biphosphate; PIP3: phosphatidylinositol 3,4,5 triphosphate; Rag: Ras-related GTPase; RHEB: Ras homolog enriched in brain; TSC: tuberous sclerosis complex). Blunt head red arrow: inhibition; sharp black arrow: stimulation.

**Figure 4 cancers-15-04043-f004:**
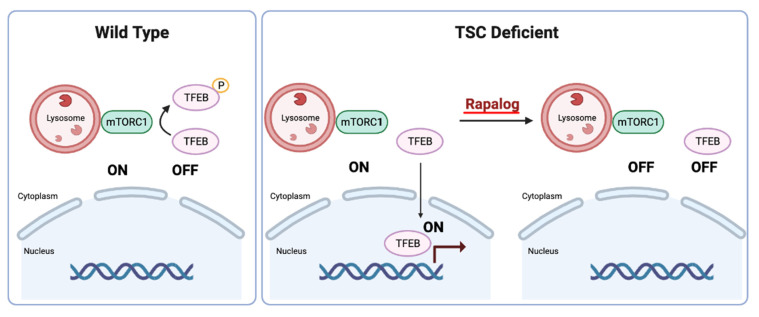
Illustration depicting TFEB regulation by mTORC1 in wild type and TSC-deficient cells (created with BioRender.com).

**Table 1 cancers-15-04043-t001:** Summary of novel renal entities with *TSC/MTOR* mutations.

Type	Clinical Features	Morphology	Immunohistochemistry	Molecular Features
**Renal cell carcinoma with fibromyomatous stroma (RCC FMS)**	Mostly sporadic and solitary, indolent	Solid, smaller tumors, tan to brown, may have lobulated appearance, clear cells with voluminous cytoplasm forming nodules, separated, and encircled by fibromuscular stroma	CK7+ CAIX+ (membranous) CD10+ AMACR−	TSC/MTOR mutations
**Eosinophilic solid and cystic renal cell carcinoma (ESC RCC)**	Mostly in females, largely sporadic and solitary, generally indolent	Solid and cystic, voluminous eosinophilic cells, cytoplasmic stippling	CK20+ CK7− CD117− Vimentin+ Cathepsin K+ (focal)	Somatic bi-allelic loss of function mutations in *TSC1* and *TSC2*
**Eosinophilic vacuolated tumor (EVT)**	Broad age range, sporadic and solitary, rare cases in TSC patients, indolent	Solid, smaller tumors, tan to brown or gray, large vessels often at the periphery, eosinophilic cells with frequent and prominent intracytoplasmic vacuoles, large nucleoli	cathepsin K+ CD117+ CD10+ CK7− (only rare cells +) CK20− Vimentin−	TSC/MTOR pathway mutations (all cases), deletions of chromosome 19 and 1 (in cases with *MTOR* mutation)
**Low-grade oncocytic tumor (LOT)**	Mostly in older patients, sporadic and solitary, rare cases in TSC patients, indolent	Solid, smaller tumors, tan to mahogany brown, sharp transition to edematous areas with scattered individual cells, round to oval nuclei without irregularities and prominent nucleoli, often perinuclear halos	CK7+ (diffuse) CD117− (rarely weak +) GATA3+ (limited data) FOXI1− CK20− Vimentin−	TSC/MTOR pathway mutations (almost all cases), lack of multiple chromosome losses, deletions of chromosome 19p, 19q, and 1p (in some cases), no *CCND1* rearrangements

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
