# Peer review of "Recent Advances in Renal Tumors with TSC/mTOR Pathway Abnormalities in Patients with Tuberous Sclerosis Complex and in the Sporadic Setting"

_cancers, 2023, doi:10.3390/cancers15164043_

Round 1

Reviewer 1 Report

I thoroughly enjoyed this review. It is very informative, showing a great depth of insight into renal tumor types and links to TSC and the mTOR pathway. This review should be of interest to the readership, and those with particular interest in renal tumors and mTOR-driven cancers. The comments below are suggestions that could help.

1)    If possible, it would be good to include something relating to HIF-1a and HIF-2a and VEGFA in angiogenic signaling and altered metabolism in RCC, which is clearly linked to mTOR signaling and TSC. Prof. James Brugarolas is an expert on HIF in TSC, so it was a surprise not to see something on this key driver of RCC.

2)    Could the authors write a few sentences on the current use of mTOR inhibitors for RCC. It is hinted at in the text, but is not actually discussed.

3)    While the authors say that mTOR inhibitors does not inhibit mTORC2, this is not altogether accurate. Everolimus is also known to partially impair mTORC2 signaling to a degree. Furthermore, allosteric binding of rapalogues to mTOR as mTOR is being translated impedes mTOR binding to mTORC2 components. Consequently, this destabilises any de novo made mTORC2 complexes over time and can lead to reduction of mTORC2 signaling also (but is not complete).

4)    The higher % of tumors in females is of interest. Do the authors think this is linked to estrogen signaling (in a similar manner to LAM).

5)    The authors mention FLCN mutations. BHD was mentioned as an abbreviation, but was not described in full. It would be worth discussing Birt-Hogg-Dube syndrome in a few sentence, where patients are at increased risk of acquiring renal tumours of all subtypes after the age of 50 (1/3 of patients will end up with renal tumors) etc. Even more important, as FLCN is again mentioned in the diagram.

6)    In the diagram, indicate that mTOR inhibition of TFE3 is via phosphorylation, and in the figure legend this stops nuclear translocation from the lysosome. It could be helpful to mention that mTORC1 resides at lysosomal membranes (translocated to lysomes by Rags when nutrients are plentiful).

Additional minor things:

1)    Check that the font size and style are the same. See sentence where font sizes are altered: ‘ Tuberous sclerosis complex (TSC) syndrome (OMIM 191100) is a multi-organ syndrome characterized by hamartomas and benign tumors in various organs [5].’ Some other sentences are larger font too.

2)    Page 3, sentence: ‘Hemorrhage is concerning particularly with actively growing AMLs that are >3 cm [29,35].

This could just be my personal preference, add >3 cm ‘in diameter’. Other uses of tumor diameter measurements in text should include the word ‘diameter’.

3)    Page 5, sentence: ‘ They occur across a broad age range, including in children, and are particularly common in females [52-54].

Add ‘ female patients with TSC’ at end of sentence.

Author Response

I thoroughly enjoyed this review. It is very informative, showing a great depth of insight into renal tumor types and links to TSC and the mTOR pathway. This review should be of interest to the readership, and those with particular interest in renal tumors and mTOR-driven cancers. The comments below are suggestions that could help.

Thank you for reviewing our manuscript.

1)    If possible, it would be good to include something relating to HIF-1a and HIF-2a and VEGFA in angiogenic signaling and altered metabolism in RCC, which is clearly linked to mTOR signaling and TSC. Prof. James Brugarolas is an expert on HIF in TSC, so it was a surprise not to see something on this key driver of RCC.

We have now added text on mTORC1 and HIF-1a signaling. Dr. Brugarolas has also made significant changes to the molecular and clinical section of the manuscript.

2)    Could the authors write a few sentences on the current use of mTOR inhibitors for RCC. It is hinted at in the text, but is not actually discussed.

We want to thank the reviewer for this suggestion. We have now added a paragraph on mTOR inhibition in RCC.

3)    While the authors say that mTOR inhibitors does not inhibit mTORC2, this is not altogether accurate. Everolimus is also known to partially impair mTORC2 signaling to a degree. Furthermore, allosteric binding of rapalogues to mTOR as mTOR is being translated impedes mTOR binding to mTORC2 components. Consequently, this destabilises any de novo made mTORC2 complexes over time and can lead to reduction of mTORC2 signaling also (but is not complete).

Thank you for this suggestion. We have removed the statement that mTOR inhibitors do not inhibit mTORC2.

4)    The higher % of tumors in females is of interest. Do the authors think this is linked to estrogen signaling (in a similar manner to LAM).

It is speculated that estrogen may have a role. We have added this to the manuscript.

5)    The authors mention FLCN mutations. BHD was mentioned as an abbreviation, but was not described in full. It would be worth discussing Birt-Hogg-Dube syndrome in a few sentence, where patients are at increased risk of acquiring renal tumours of all subtypes after the age of 50 (1/3 of patients will end up with renal tumors) etc. Even more important, as FLCN is again mentioned in the diagram.

We have clarified the abbreviation, however, a more detailed discussion on BHD seems a bit beyond the scope of this review article.

6)    In the diagram, indicate that mTOR inhibition of TFE3 is via phosphorylation, and in the figure legend this stops nuclear translocation from the lysosome. It could be helpful to mention that mTORC1 resides at lysosomal membranes (translocated to lysomes by Rags when nutrients are plentiful).

We have made the suggested modifications to the figure and legend.

Additional minor things:

  • Check that the font size and style are the same. See sentence where font sizes are altered: ‘ Tuberous sclerosis complex (TSC) syndrome (OMIM 191100) is a multi-organ syndrome characterized by hamartomas and benign tumors in various organs [5].’ Some other sentences are larger font too.

Thank you. We have made the font uniform.

  • Page 3, sentence: ‘Hemorrhage is concerning particularly with actively growing AMLs that are >3 cm [29,35]. This could just be my personal preference, add >3 cm ‘in diameter’. Other uses of tumor diameter measurements in text should include the word ‘diameter’.

We have added the work “diameter” throughout the text.

  • Page 5, sentence: ‘ They occur across a broad age range, including in children, and are particularly common in females [52-54]. Add ‘ female patients with TSC’ at end of sentence.

Done.

Reviewer 2 Report

Kapur et al.

This is a comprehensive and beautifully written review of TSC-associated RCC by an expert team.  The work is timely and important since the WHO has recently reclassified some forms of RCC that are associated with TSC/mTOR pathway mutations.  The most recent review of this topic was in 2021 PMID: 34680979, prior to the new WHO guidelines.  This is a rapidly emerging area, and the authors have captured the many moving parts with exceptional clarity.  This will be very useful to the RCC and TSC communities from clinical and research perspectives.  There are only two important concerns, which are detailed below.  The first is the use of “tumor” instead of RCC and the second is related to the sections in which clinical management is discussed. 

1)    In the section on angiomyolipomas and variants, the epithelioid AML discussion could be more in-depth, especially since this is an area of considerable confusion for clinicians.  It would be helpful to say more about the distinguishing features of the epithelioid AML, including the fact that these are primarily sporadic and rarely occur in TSC, and the clarify the terms PEComa and malignant PEComa and their relationship to epithelioid AML.  From the pathology perspective how precisely are these terms defined?

2)    The section about the management of renal lesions could be expanded and strengthened.  The results of the pivotal clinical trials for AML (including the Exist trial) should be discussed, including the long-term benefit and the regrowth upon treatment discontinuation. 

3)    Elective embolization of growing AML is not mentioned as a treatment option.  This is an option for sporadic AML and occasionally, in specific situations, for TSC AML. 

4)    In the same paragraph, the authors state that “identification of a small renal tumor in the setting of multifocal AML requires careful surgical planning to not leave the patient anephric” should be stated with more nuance – do the authors mean a small biopsy-proven RCC?  How small?  Since a major or perhaps THE major cause of renal disease in TSC is surgical resection of tumors, any mention of surgery should be discussed with great care.  Many small renal tumors should be observed or biopsied, not resected, in TSC.  The management of small renal tumors in the general population often differs from management in TSC, and many TSC patients are treated in the community instead of in academic specialized settings.  This article is likely to have a broad reach, hence it could be important to provide more context. 

5)    Biopsy plays an important role in the diagnosis of growing, non-fat containing renal lesions in TSC.  This should be discussed. 

6)    The authors conclude this paragraph with a case report of a TSC patient with LOT that benefited from mTORC1 inhibition, but LOT are not malignant, so it is confusing that this is in the same paragraph as a discussion of the management of RCC.  Treatment with mTORC1 inhibition is not a standard of care for TSC-RCC and perhaps this should be specified. 

7)    In lieu or in addition to addressing the concerns above related to clinical management, the authors might want to include a specific statement that treatment of TSC-associated renal tumors often requires specialized multi-disciplinary expertise and that clinical management has recently been reviewed elsewhere and cite a paper such as PMID: 33006051. 

8)    On page 3, the authors introduce TSC-RCC, but refer to these as “tumors.” “Renal tumors occur in 2-4% of TSC patients” – but renal tumors (including AML) occur in up to 85% of TSC patients.  It seems clear that the authors are discussing TSC-RCC here and it’s not evident why the term RCC is not used. 

9)    It is important to emphasize that RCC in TSC occur at a younger age than the general population, including in children.  Also, some TSC patients develop multiple, bilateral TSC (for example reference #13).  The reasons for this patient-specific susceptibility to TSC-RCC is not understood.  Because of these considerations, the comment that the incidence is “slightly higher than in the general population” might be misleading to some readers. 

Author Response

This is a comprehensive and beautifully written review of TSC-associated RCC by an expert team.  The work is timely and important since the WHO has recently reclassified some forms of RCC that are associated with TSC/mTOR pathway mutations.  The most recent review of this topic was in 2021 PMID: 34680979, prior to the new WHO guidelines.  This is a rapidly emerging area, and the authors have captured the many moving parts with exceptional clarity.  This will be very useful to the RCC and TSC communities from clinical and research perspectives.  There are only two important concerns, which are detailed below.  The first is the use of “tumor” instead of RCC and the second is related to the sections in which clinical management is discussed. 

Thank you for reviewing our manuscript.

1)    In the section on angiomyolipomas and variants, the epithelioid AML discussion could be more in-depth, especially since this is an area of considerable confusion for clinicians.  It would be helpful to say more about the distinguishing features of the epithelioid AML, including the fact that these are primarily sporadic and rarely occur in TSC, and the clarify the terms PEComa and malignant PEComa and their relationship to epithelioid AML.  From the pathology perspective how precisely are these terms defined?

We have added a discussion related to PEComa and distinguishing features of epithelioid AML.

2)    The section about the management of renal lesions could be expanded and strengthened.  The results of the pivotal clinical trials for AML (including the Exist trial) should be discussed, including the long-term benefit and the regrowth upon treatment discontinuation. 

Thank you for this suggestion. We have expanded the section on management.

3)    Elective embolization of growing AML is not mentioned as a treatment option.  This is an option for sporadic AML and occasionally, in specific situations, for TSC AML. 

We have added this statement.

4)    In the same paragraph, the authors state that “identification of a small renal tumor in the setting of multifocal AML requires careful surgical planning to not leave the patient anephric” should be stated with more nuance – do the authors mean a small biopsy-proven RCC?  How small?  Since a major or perhaps THE major cause of renal disease in TSC is surgical resection of tumors, any mention of surgery should be discussed with great care.  Many small renal tumors should be observed or biopsied, not resected, in TSC.  The management of small renal tumors in the general population often differs from management in TSC, and many TSC patients are treated in the community instead of in academic specialized settings.  This article is likely to have a broad reach, hence it could be important to provide more context. 

We have clarified these points.

5)    Biopsy plays an important role in the diagnosis of growing, non-fat containing renal lesions in TSC.  This should be discussed. 

We have further clarified this statement.

6)    The authors conclude this paragraph with a case report of a TSC patient with LOT that benefited from mTORC1 inhibition, but LOT are not malignant, so it is confusing that this is in the same paragraph as a discussion of the management of RCC.  Treatment with mTORC1 inhibition is not a standard of care for TSC-RCC and perhaps this should be specified. 

We have moved this line before the discussion on RCC treatment.

7)    In lieu or in addition to addressing the concerns above related to clinical management, the authors might want to include a specific statement that treatment of TSC-associated renal tumors often requires specialized multi-disciplinary expertise and that clinical management has recently been reviewed elsewhere and cite a paper such as PMID: 33006051. 

We have added this reference.

8)    On page 3, the authors introduce TSC-RCC, but refer to these as “tumors.” “Renal tumors occur in 2-4% of TSC patients” – but renal tumors (including AML) occur in up to 85% of TSC patients.  It seems clear that the authors are discussing TSC-RCC here and it’s not evident why the term RCC is not used. 

We have changed “Renal tumors” to “Renal cell carcinoma”.

9)    It is important to emphasize that RCC in TSC occur at a younger age than the general population, including in children.  Also, some TSC patients develop multiple, bilateral TSC (for example reference #13).  The reasons for this patient-specific susceptibility to TSC-RCC is not understood.  Because of these considerations, the comment that the incidence is “slightly higher than in the general population” might be misleading to some readers. 

Thank you for this suggestion. We have modified this text.